# Attenuation of Pulmonary Fibrosis by the MyD88 Inhibitor TJ-M2010-5 Through Autophagy Induction in Mice

**DOI:** 10.3390/biomedicines13092214

**Published:** 2025-09-10

**Authors:** Yang Yang, Zeyang Li, Minghui Zhao, Yuanyuan Zhao, Zhimiao Zou, Yalong Xie, Limin Zhang, Dunfeng Du, Ping Zhou

**Affiliations:** Key Laboratory of Organ Transplantation, Ministry of Education, NHC Key Laboratory of Organ Transplantation, Key Laboratory of Organ Transplantation, Chinese Academy of Medical Sciences, Institute of Organ Transplantation, Tongji Hospital, Tongji Medical College, Huazhong University of Science and Technology, Wuhan 430030, China; yangya1901@163.com (Y.Y.); zeyangli1995@hust.edu.cn (Z.L.); m202176260@hust.edu.cn (M.Z.); yyzhao@tjh.tjmu.edu.cn (Y.Z.); zouzhimiao@hust.edu.cn (Z.Z.); xieyalong2013@163.com (Y.X.)

**Keywords:** idiopathic pulmonary fibrosis, myeloid differentiation factor 88, autophagy, lung fibroblast, TJ-M2010-5

## Abstract

**Background and Objectives:** Idiopathic pulmonary fibrosis (IPF) is a fatal interstitial lung disease with few effective treatments. In its pathogenesis, damage-associated molecular patterns are released and recognized by Toll-like receptors (TLRs); all TLRs except TLR3 transduce signals through MyD88. Research has shown that autophagy participates in the progression of pulmonary fibrosis, and MyD88 is closely associated with autophagy. However, whether targeting MyD88 can affect fibrosis progression by regulating autophagy during lung fibrosis remains unclear. **Materials and Methods:** TJ-M2010-5 (TJ-5) is a small molecular derivative of aminothiazole that inhibits MyD88 homodimerization. A bleomycin-induced pulmonary fibrosis model in mice was established, and a human lung fibroblast cell line MRC-5 was cultured, and the mechanism of fibrosis induced by TGF-β1 was studied. TJ-5 and the autophagy inhibitor 3-MA were used to intervene. **Results:** Our study indicated that TJ-5 suppressed fibrosis foci formation and collagen deposition in fibrotic lungs, effectively increased the survival rate of bleomycin-stimulated mice from 40.0% to 80.0%, and repressed lung fibroblast activation in vitro. Subsequently, TJ-5 could trigger autophagy, as indicated by increased autophagosomes, LC3B-II and Beclin-1 promotion, and p62 degradation. Moreover, inhibition of TJ-5-induced autophagy by 3-MA reversed the anti-fibrosis effect of TJ-5. Furthermore, the autophagy-related pathways PI3K/AKT/mTOR and MAPK/mTOR were inhibited under TJ-5 intervention. **Conclusions:** Our findings demonstrated that the mechanism of TJ-5 in alleviating lung fibrosis was through triggering MyD88-related autophagy, and TJ-5 may be therapeutically useful for the clinical treatment of IPF.

## 1. Introduction

Idiopathic pulmonary fibrosis (IPF) is a destructive and progressive interstitial lung disease that ultimately leads to respiratory failure [1]. It has a very poor prognosis, with a median survival of only 3–5 years, which is worse than that of some severe cancers [2]. The global incidence and prevalence of IPF are estimated to be in the range of 0.09–1.30 and 0.33–4.51 per 10,000 persons, respectively [3]. The development of pulmonary fibrosis is characterized by the differentiation of fibroblasts to myofibroblasts [4,5]. Myofibroblasts expand excessively and secrete excessive amounts of accumulated extracellular matrix (ECM) proteins, leading to lung architecture dysregulation and the progressive loss of respiratory function [6,7,8]. However, few drugs can be clinically applied; therefore, effective intervention drugs for IPF are urgently needed.

Autophagy is an evolutionarily conserved intracellular degradation process whereby impaired organelles or proteins are removed, maintaining cellular homeostasis [9]. Impaired autophagy has been implicated in the pathogenesis of IPF [10,11,12]. Autophagic deficiency in IPF fibroblasts promotes its activation and the production of ECM proteins [13,14,15]; Beclin-1 (autophagic gene) deficiency leads to collagen deposition in the fibroblasts of IPF patients [16]. Moreover, stimulation by the pro-fibrotic factor TGF-β1 can inhibit autophagy in lung fibroblasts in vitro [17].

In the pathogenesis of IPF, external injury causes damaged and activated cells to release damage-associated molecular patterns that are recognized by Toll-like receptors (TLRs), a group of highly conserved pattern recognition receptors [18]. Among all TLRs, TLR2 [19], TLR4 [20], and TLR9 [21] are essential for lung fibrogenesis; however, they transduce signals through myeloid differentiation primary factor 88 (MyD88), a common adaptor molecule. The study indicated that MyD88-deficient mice showed a reduction of fibrosis in interstitial lung disease [22]. Meanwhile, several studies have found that MyD88 is closely associated with autophagy [23,24,25]. Kader et al. demonstrated that MyD88-dependent autophagy inhibition aggravated Ehrlichia-induced liver injury [26]. As mentioned above, autophagy is a key pathophysiological player in pulmonary fibrosis. However, whether inhibition of MyD88 affects autophagy activity during pulmonary fibrosis remains unclear.

Autophagy can be regulated by multiple signaling pathways and effectors [27]. The mammalian target of rapamycin (mTOR) was considered the core negative regulator of autophagy [28], which could be regulated by the primary autophagy regulatory pathway, the PI3K (phosphoinositide 3-kinase)/AKT pathway [29], and the mitogen-activated protein kinase (MAPK) subfamilies, such as p38, extracellular signal-regulated kinase (ERK), and c-Jun N-terminal kinase (JNK) [30,31,32]. Meanwhile, those signaling pathways play vital roles in the development and progression of lung fibrosis. Whereas PI3K/AKT and MAPK pathways are downstream of TLR/MyD88 [33,34], therefore, we hypothesize that TLR/MyD88 regulates autophagy in IPF through PI3K/AKT/mTOR and MAPK/mTOR signaling pathways.

Recently, we synthesized the MyD88 inhibitor, TJ-M2010-5 (abbreviated as TJ-5), an aminothiazole derivative, and inhibited MyD88 homodimerization. This result was shown in Figure 1A in our previous study [35]. In this study, we used TJ-5 as an effective MyD88 inhibitor; therefore, we investigated the therapeutic effects of TJ-5 on pulmonary fibrosis in mice and elucidated the impacts on autophagy.

## 2. Materials and Methods

### 2.1. Reagents

TJ-5, known as 3-(4-(4-benzylpiperazin-1-yl)-N-(4-phenylthiazol-2-yl)) propanamide, a targeted inhibitor of MyD88 dimerization, was synthesized at the Academy of Pharmacy, Tongji Medical College, Huazhong University of Science and Technology, Wuhan, China (WIPO Patent Application Number: PCT/CN2012/070811). It was dissolved in distilled deionized water (ddH_2_O). The structure of TJ-5 is presented in Figure 1A, and the details have been reported in our previous article [35]. BLM was purchased from Tongji Hospital, transforming growth factor-β1 (TGF-β1) was purchased from PeproTech (Rochy Hill, NJ, USA), and 3-methyladenine (3-MA) was purchased from Selleck (Shanghai, China).

### 2.2. Animals and Animal Treatment

Wild-type C57BL/6J mice (male, 8–10 weeks old) were purchased from Weitonglihua Company (Beijing, China). Mice were raised under standard controlled conditions (21 ± 2 °C, 55 ± 10% humidity, and 12 h light/dark cycle) and in a specific pathogen-free environment. Mice were allowed free access to food and water. They were acclimatized to the environment for at least 7 days before the experiments were performed. All animal experiments were approved by the Animal Ethics Committee of Tongji Hospital (ethical approval number: TJH-201901008).

For the BLM-induced pulmonary fibrosis model, mice were randomly assigned to three groups: the normal saline (NS) group, the BLM group, and the BLM + TJ-5 group. A total of 54 mice were used, with 6 mice in each group. After 8 h of overnight fasting, mice were anesthetized briefly using intraperitoneal injection of sodium pentobarbital, and pulmonary fibrosis was induced with a single dose of BLM (2 mg/kg, dissolving in sterile saline) via intratracheal instillation on day 0. The mice in the NS group received equal volumes of sterile saline. Mice in the TJ-5 group were subjected to intraperitoneal injections of TJ-5 (30 mg/kg) once daily, while mice in the NS group and BLM group were injected with an equal volume of ddH_2_O. The mice with the prophylactic intervention were injected with TJ-5 once daily for 2 days before the BLM challenge. The mice with the therapeutic intervention were injected with TJ-5 at D7 after BLM challenge. Lung tissues were collected 21 days after BLM injection. The dose of TJ-5 administration was based on previous research [35].

For the BLM-induced acute lung injury model, the groupings of mice and modeling methods were the same as those used for the BLM-induced pulmonary fibrosis model, with 5 mice in each group. Samples were harvested after 3 days and 7 days of treatment with TJ-5. Mice were sacrificed by cervical dislocation under deep anesthesia at the end of the experiment.

### 2.3. Histology and Immunohistochemistry

The left lobe of the mouse lung was fixed with fresh 4% paraformaldehyde for 24 h and embedded in paraffin. Then, 4 μm sections were stained with hematoxylin and eosin (H&E) and Masson’s trichrome. The histologic sections were examined and photographed using light microscopy. Lung injury scores were evaluated using a previously described method [36], and the severity of fibrosis was quantified by the modified Ashcroft score [37]. For immunohistochemical staining, the lung sections were incubated with primary antibodies against α-SMA (#19245, CST, Beverly, MA, USA), fibronectin (FN, sc-8422, Santa Cruz Biotechnology, Santa Cruz, CA, USA), and COL1A1 (A16891, Abclonal, Wuhan, China); a detailed description is provided elsewhere [38].

### 2.4. Bronchoalveolar Lavage Fluid Collection and the Number of Leukocytes in Bronchoalveolar Lavage Fluid

The mice were anesthetized on days 3 and 7 after BLM stimulation, and a 22-gauge catheter was inserted into the trachea. Bronchoalveolar lavage fluid (BALF) was collected using 1 mL sterile saline through catheter washing three times. Approximately 1.8 mL of BALF was recovered from each mouse and centrifuged at 8000 rpm for 5 min at 4 °C. The cell pellet was prepared for cell counting, and the supernatant was prepared to measure the protein concentration. Cells were resuspended in sterile phosphate-buffered saline containing 2% fetal bovine serum, and the total number of cells was counted using a hemocytometer. Then, differential leukocytes were counted on cytospin slides stained with Diff-Quick.

### 2.5. Measurements of Contents of Hydroxyproline and Activity of Superoxide Dismutase in Lung Tissues

At 21 days after BLM treatment, lung tissues were removed. Hydroxyproline (HYP) contents and myeloperoxidase (MPO) activity in the lung tissue homogenate were measured using HYP kits (BioVision, Milpitas, CA, USA) and MPO kits (Jiancheng Biotech, Nanjing, China) according to the instructions provided by the manufacturers.

### 2.6. ELISA Assay

The concentrations of interleukin-1β (IL-1β), IL-6, tumor necrosis factor-α (TNF-α), monocyte chemoattractant protein-1 (MCP-1), and TGF-β1 in BALF and lung homogenate were measured using a corresponding ELISA kit (Dakewe, Shenzhen, China) according to the instructions provided by the manufacturer.

### 2.7. Real-Time Polymerase Chain Reaction Analysis

Total RNA was isolated with TRIzol reagent (Takara, Tokyo, Japan) and reverse-transcribed to cDNA with an RNA simple Total RNA Kit (Takara). A real-time polymerase chain reaction (RT-PCR) was performed using Hieff^TM^ qPCR SYBR^®^ Green Master MixI (High Rox Plus; Yesen, Shanghai, China). Results were normalized using GAPDH expression. Detailed primer sequences for MyD88, IL-1β, IL-6, TNF-α, MCP-1, TGF-β1, α-SMA, FN, and COL1A1 are listed in Appendix A.

### 2.8. Cell Culture and Drug Interventions

Human lung fibroblast MRC-5 cells were purchased from the Cell Bank of Chinese Academy of Sciences (Shanghai, China) and used, ranging from passage 27 to 37. Cells were cultured in Eagle’s minimal essential medium (Thermo Fisher Scientific, Waltham, MA, USA) supplemented with 10% fetal bovine serum (Thermo Fisher Scientific), 1% non-essential amino acids (Thermo Fisher Scientific), 1% Gluta-max (Thermo Fisher Scientific), 1% sodium pyruvate (Thermo Fisher Scientific), and 1% penicillin/streptomycin (Thermo Fisher Scientific). Cells were kept at 37 °C in a 5% CO_2_ humidified atmosphere. Cells of the TJ-5 group were pretreated with TJ-5 at varying concentrations (10 μM, 15 μM, 20 μM, 30 μM) for 2 h before stimulation with TGF-β1 (5 ng/mL) unless otherwise indicated; 72 h later, the cells were collected for subsequent experiments. For certain experiments, the cells were also co-treated with 3-MA.

### 2.9. Cell Immunofluorescence

MRC-5 cells were cultured on 12-well chamber slides and fixed with 4% paraformaldehyde after the intervention. Then, they were stained with anti-LC3B (A19665; ABclonal) monoclonal antibody overnight at 4 °C. After washing, the slides were incubated with goat anti-rabbit FITC-conjugated antibody in the dark for 1 h at 37 °C and counterstained with DAPI for 10 min at 20 °C. Then, the slides were visualized using a fluorescence microscope (Olympus, Tokyo, Japan).

### 2.10. Proliferation, Apoptosis, and Cytotoxicity Assays

Cell proliferation was measured using cell counting kit-8 (CCK-8; Dojindo Molecular Technologies, Kannaimachi, Japan). MRC-5 cells were seeded at 1 × 10^4^ cells/well in 96-well plates and cultured overnight; then, they were supplemented with TJ-5 at concentrations of 0 μM, 10 μM, 15 μM, 20 μM, and 30 μM at 37 °C. After 72 h, the media were replaced by fresh media and CCK-8 comprising 90 μL media and 10 μL CCK-8 per well; then, the cells were incubated for 1 h at 37 °C. Absorbance at 450 nm was detected using a microplate reader. MRC-5 cell apoptosis was quantified using the Annexin V/PI apoptosis detection kit (Becton Dickinson, Franklin Lakes, NJ, USA) according to the instructions provided by the manufacturer. TJ-5 was added (10 μM, 15 μM, 20 μM, 30 μM) to the culture for 40 min, followed by the addition of TGF-β1 (5 ng/mL) for 72 h; then, MRC-5 cells were measured using a fluorescence-activated cell sorter analysis, and the results were analyzed using FlowJo software (version 10.6.2). Cellular toxicity was detected using the lactate dehydrogenase (LDH) cytotoxicity assay kit (Beyotime Biotechnology, Shanghai, China) according to the instructions provided by the manufacturer. The procedure by which drugs were added to MRC-5 cells was the same as that described for apoptosis detection.

### 2.11. Western Blotting

The total protein was extracted from lung tissues and cultured cells (MRC-5 cells), respectively, according to the instructions provided by the manufacturer (Beyotime Biotechnology). The same batch of protein samples was electrophoresed on different SDS-PAGE gels at the same time, respectively. After separation on SDS-PAGE, the proteins were transferred onto the PVDF membrane, and the PVDF membrane was cut according to the molecular weight of the target protein. Immediately after, the proteins were blocked with 5% nonfat milk for 2 h at 20 °C. Then, proteins were incubated with the indicated primary antibodies overnight at 4 °C. The membranes were washed with 1×TBST four times and incubated with HRP-Goat anti-rabbit/mouse secondary antibodies (ABclonal) for 2 h at 20 °C. Thereafter, the membranes were washed four times and visualized with enhanced chemiluminescence (Beyotime Biotechnology). The primary antibodies used included: α-SMA (#19245, CST), FN (sc-8422, Santa Cruz Biotechnology), COL1A1 (#84336, CST), MyD88 (#4283, CST), TLR2 (#13744, CST), TLR4 (ab13867, Abcam, Cambridge, MA, USA), TLR9 (A14642, ABclonal), TGF-β1 (21898-1-AP, Proteintech, Wuhan, China), p-Smad2/3 (#8828, CST), Smad2/3 (#8685, CST), p-p38 (#4511, CST), p38 (#8690, CST), p-ERK (#4376, CST), ERK (#4695, CST), p-JNK (#4668, CST), JNK (#9252, CST), LC3B (A19665, ABclonal), SQSTM1/p62 (A19700, ABclonal), Beclin-1 (A7353, ABclonal), p-mTOR (AP0115, ABclonal), mTOR (A2445, ABclonal), p-PI3K (ab182651, Abcam), PI3K (#4292, CST), p-AKT (#4060, CST), AKT (#9272, CST), and GAPDH (ab37168, Abcam).

### 2.12. Transmission Electron Microscopy

Lung tissues were collected on day 21 after BLM stimulation, immediately fixed with 2.5% glutaraldehyde for 2 h at 4 °C, and post-fixed with 2% osmium tetroxide for 1.5 h at 20 °C. Then, transmission electron microscopy samples were embedded in 1 × 1 × 1 mm lung tissue blocks, and ultra-thin sections (80–100 nm) were cut from the lung tissue. The ultra-thin sections were observed by transmission electron microscopy at 80 kV after staining with uranyl acetate/lead citrate.

### 2.13. Statistical Analysis

All statistical analyses were performed using GraphPad Prism (version 6.02) software. All data are presented as means ± SEM from at least three independent experiments. Paired comparisons were assessed using Student’s *t*-test, and multiple group comparisons were analyzed using a one-way analysis of variance followed by Dunnett’s test. Survival rates were compared using a log-rank test. *p* < 0.05 was considered statistically significant.

## 3. Results

### 3.1. MyD88 Expression Was Markedly Upregulated in Fibrotic Lungs of BLM-Induced Mice and TGF-β1-Stimulated Fibroblasts

Lung tissues were obtained at 3, 7, and 21 days after the BLM challenge. The MyD88 mRNA level was dramatically increased after the BLM injection (Figure 1B). Moreover, the protein expression level of MyD88 was substantially upregulated in the BLM group compared with the NS group (Figure 1C,D). We further confirmed that the MyD88 protein level was also markedly increased in MRC-5 human lung fibroblasts with TGF-β1 intervention (Figure 1E,F). Our results showed that MyD88 expression was positively correlated with pulmonary fibrosis and lung fibroblast activation.

### 3.2. TJ-5 Ameliorated BLM-Induced Pulmonary Fibrosis and Reduced the Expression of Fibrosis Markers

Mice were euthanized 21 days after BLM injection, and the effects of TJ-5 were examined in lung tissues. As indicated in Figure 2A, lung inflammation and fibrosis were evaluated by H&E and Masson’s trichrome, and treatment with TJ-5 significantly suppressed fibrosis foci formation and collagen deposition in fibrotic lungs compared with lung injury caused by BLM. Fibrosis severity was estimated using the Ashcroft score (Figure 2B). Furthermore, the effects of TJ-5 on collagen accumulation were ultimately demonstrated by lower HYP levels in the lungs (Figure 2C). TJ-5 administration in the lungs markedly improved the survival rate from 40.0% to 80.0% at 28 days (Figure 2D) and abated the weight loss caused by BLM (Figure 2E).

ECM proteins, like α-SMA, FN, and COL1A1, are important fibrotic markers during the fibrotic progression of pulmonary fibrosis [39]. To confirm whether TJ-5 can affect the transcription and expression of ECM genes, we measured the mRNA and protein levels of ECM genes in fibrotic lung tissues. As shown in Figure 2F, the α-SMA, FN, and COL1A1 mRNA levels were significantly higher in the BLM group than in the NS group. Strikingly, the administration of TJ-5 efficiently prevented an increase in these mRNA levels. Moreover, the TJ-5 intervention also dramatically inhibited BLM-induced ECM protein expression compared with the BLM group (Figure 2G–J).

Additionally, the TGF-β1/Smad2/3 signaling pathway is also a major profibrotic factor. As shown in Figure 2K, the TGF-β1 mRNA expression was increased in the BLM group and significantly reduced after treatment with TJ-5. Moreover, TGF-β1 and phosphorylation of Smad2/3 were significantly increased in mice; however, TJ-5 intervention led to a sharp decline in the protein expression of TGF-β1 and p-Smad2/3 (Figure 2L,M).

These data suggested that TJ-5 could effectively inhibit ECM protein expression and activate TGF-β1/Smad2/3 signaling.

### 3.3. TJ-5 Attenuated BLM-Induced Pulmonary Inflammation

BLM exposure used to induce IPF in mice could increase tissue-infiltrating proinflammatory cells, cytokines, and chemokines, leading to fibrosis [40]. We evaluated whether the TJ-5 reduction of fibrosis was attributable to the inhibition of early inflammation stages. The lung histological changes caused by BLM-induced acute lung injury at 3 and 7 days were analyzed by H&E staining (Figure 3A). The alveolar septum was markedly thickened, and inflammatory cell infiltration and pulmonary edema were aggravated in the BLM group (Figure 3A,B); however, the administration of TJ-5 effectively alleviated these symptoms. Moreover, TJ-5 reduced the total BALF protein concentration (Figure 3C); the counts of total cell number (Figure 3D), macrophages (Figure 3E), and neutrophils (Figure 3F) in BALF; and the MPO activity (Figure 3G). These markers’ levels can indicate the severity of pulmonary inflammation. Furthermore, as compared with the NS group, the RT-PCR analyses of lung tissues showed a strong upregulation of IL-1β, IL-6, TNF-α, and MCP-1 on days 3 (Figure 3H) and 7 (Figure 3I). However, TJ-5 significantly inhibited the increases in pulmonary IL-1β, IL-6, TNF-α, MCP-1, and TGF-β1 (Figure 3H,I). The ELISA analyses of BALF and lung homogenate also indicated a decrease in levels of IL-1β, IL-6, TNF-α, MCP-1, and TGF-β1 in the TJ-5 group compared with the BLM group on days 3 (Figure 3J,L) and 7 (Figure 3K,M). These results suggested that TJ-5 effectively alleviated BLM-induced inflammation during the early stage of lung fibrosis.

### 3.4. TJ-5 Suppressed TGF-β1-Induced ECM Deposition in Lung Fibroblasts and Inhibited Their Proliferation

Activated fibroblasts producing ECM proteins and the aggregation of fibroblasts represent key steps in pulmonary fibrosis [41]. First, to assess the cytotoxicity of TJ-5, we measured LDH release in culture supernatants of MRC-5 cells treated with varying concentrations of TJ-5 followed by TGF-β1 stimulation; no cytotoxicity was observed (Figure 4A). Then, we examined the effects of preintervention using TJ-5 on fibroblast activation with subsequent TGF-β1 stimulation during an immunoblot analysis. TGF-β1 largely induced the production of α-SMA, FN, and COL1A1 (Figure 4B,C); however, TJ-5 effectively suppressed the expression of these proteins in a dose-dependent manner. Moreover, the CCK-8 assay was performed to detect the inhibitory effects of TJ-5 on MRC-5 cell proliferation. The results indicated that TJ-5 could significantly inhibit MRC-5 cell proliferation in a dose-dependent manner (Figure 4D). Meanwhile, we measured the apoptosis of TGF-β1-induced MRC-5 cells using Annexin V/PI methods and found that it slowly and gradually increased as the concentration of TJ-5 increased, but without statistical significance (Figure 4E). Furthermore, we detected the expressions of cleaved caspase 3 and Bcl-2, and found that TJ-5 at 20 μM and 30 μM markedly downregulated cleaved caspase-3 levels, while upregulating Bcl-2 (Appendix A). Our results confirmed that TJ-5 promoted lung fibroblast apoptosis and inhibited its activation and proliferation.

### 3.5. TJ-5 Induced Autophagy in Fibrotic Lungs of BLM-Stimulated Mice and TGF-β1-Induced Fibroblasts

Autophagy can degrade aberrant proteins and is vital in the pathogenesis of pulmonary fibrosis [42]. We evaluated the effects of TJ-5 on autophagy in lung tissues and TGF-β1-induced fibroblasts through autophagosome formation, Western blot analysis, and immunofluorescence staining. Transmission electron microscopy showed higher autophagosomes in fibrotic lungs of the TJ-5 group than in the BLM group (Figure 5A,B). Similarly, in vivo results indicated that autophagic activity was markedly inhibited in the BLM group, as evidenced by the decreased expression of LC3B-II and Beclin-1 and increased level of p62 in the BLM group compared with the NS group; after TJ-5 administration, autophagic activity was significantly enhanced (Figure 5C). Moreover, immunofluorescence detection in vitro showed that TJ-5 largely increased punctate LC3B staining compared with treatment with TGF-β1 alone in MRC-5 cells (Figure 5D). Then, autophagic proteins were examined in MRC-5 cells. It was found that TJ-5 promoted the expression of LC3B-II and Beclin-1, and the degradation of p62 compared with the TGF-β1group (Figure 5E).

### 3.6. Inhibition of Autophagy Reversed the Protective Effects of TJ-5

To further verify that TJ-5 could reduce fibrosis by inducing autophagy, mice were treated with the autophagy inhibitor 3-MA. As shown in Figure 6A,B, we observed that 3-MA aggravated the extent of fibrosis in the presence of TJ-5 and caused more severe lung architecture destruction than TJ-5 alone in the lung tissue sections. In addition, the levels of ECM proteins in the BLM + TJ-5 + 3-MA group were significantly higher than those in the BLM + TJ-5 group (Figure 6C,D). These results indicated that 3-MA could reverse the in vivo effect of TJ-5.

Moreover, 3-MA was applied to MRC-5 cells in vitro. We found that 3-MA effectively reversed the inhibitory effect of TJ-5 on proliferation in TGF-β1-stimulated MRC-5 cells (Figure 6E). More importantly, under the intervention of 3-MA, TJ-5-induced LC3B-II and Beclin-1 promotion (Figure 6F,H), p62 degradation (Figure 6F,H), and ECM protein decrease (Figure 6G,I) were noticeably reversed. These data suggested that inhibition of autophagy might promote the proliferation and activation of lung fibroblasts and aggravate the expression of ECM proteins in vitro.

### 3.7. Therapeutic Intervention with TJ-5 Ameliorated BLM-Induced Pulmonary Fibrosis and Inhibited Lung Fibroblast Activation

In addition to prophylactic intervention, we tried to start dosing from D7 and tested key indicators. As indicated in Figure 7A–C, TJ-5 suppressed fibrosis foci formation and collagen deposition in BLM-stimulated lungs. Meanwhile, in the presence of TJ-5, the BLM-induced ECM protein expression was inhibited (Figure 7D,E) and autophagy activity was promoted (Figure 7F,G) compared with the BLM group. The results indicated that TJ-5 administration from D7 to D21 can also alleviate pulmonary fibrosis in mice. This implies that TJ-5 can inhibit BLM-induced fibrosis in the later stage.

As shown in Figure 8, when MRC-5 fibroblasts were first stimulated by TGF-β1 for 4 h, followed by TJ-5 stimulation, TJ-5 also significantly suppressed the expression of ECM proteins in a dose-dependent manner (Figure 8A,B). Moreover, it was also found that post-treatment with TJ-5 increased the protein level of LC3B-II and Beclin-1 and decreased that of p62 compared with the TGF-β1 group (Figure 8C,D). The results indicated that TJ-5 intervention following TGF-β1 stimulation could also inhibit lung fibroblast activation and enhance the autophagy level in lung fibroblasts.

### 3.8. TJ-5 Triggered Autophagy Possibly by Suppressing PI3K/AKT/mTOR and MAPK/mTOR Pathways

mTOR is a major negative regulator of autophagy [43]. It has been proven that TJ-5 could activate autophagy; therefore, we needed to confirm whether TJ-5-mediated autophagy is related to the mTOR activation. Hence, we measured the phosphorylation of mTOR in BLM-stimulated lung tissue and TGF-β1-challenged MRC-5 cells and found that the levels of phosphorylated mTOR in the BLM (Figure 9A,E) and TGF-β1 (Figure 9B,F) groups were dramatically higher than those in the NS and control groups, respectively. After TJ-5 intervention, mTOR phosphorylation levels were substantially decreased in the TJ-5 group in vivo (Figure 9A,E) and in vitro (Figure 9B,F).

Recent studies have shown that the PI3K/AKT and MAPK pathways can regulate autophagy by affecting mTOR activity [30,44]. Meanwhile, PI3K/AKT and MAPK pathways are downstream of TLR/MyD88 signals. Therefore, we attempted to determine whether they regulate autophagy via MyD88. First, we measured the in vivo effects of TJ-5 on the PI3K/AKT and MAPK pathways. The phosphorylation of PI3K, AKT, p38, ERK, and JNK was increased in the BLM group, and TJ-5 significantly downregulated the phosphorylation of the PI3K/AKT and MAPK pathways (Figure 9C,G). Next, during the in vitro study, the phosphorylation of the PI3K/AKT and MAPK pathways was increased in activated MRC-5 cells; however, the phosphorylation of PI3K, AKT, p38, ERK, and JNK was almost blocked as TJ-5 intervention concentrations increased (Figure 9D,H). The in vivo results were concordant with the in vitro results. Accordingly, these results suggested that TJ-5 might regulate autophagy through the PI3K/AKT/mTOR and MAPK/mTOR signaling pathways.

## 4. Discussion

Our results demonstrated for the first time that TJ-5 displayed a strong protective role toward BLM-stimulated pulmonary fibrosis in mice and activation of lung fibroblasts. Moreover, inhibition of MyD88 with TJ-5 induced autophagy, and inhibition of autophagy by 3-MA reversed the anti-fibrotic effect of TJ-5. Furthermore, we identified that TJ-5 may induce autophagy through suppression of PI3K/AKT/mTOR and MAPK/mTOR pathways. Collectively, our research suggests that pharmacologic targeting of MyD88 may provide a novel anti-fibrosis method, and TJ-5 may be a promising candidate for clinical treatment of IPF.

The importance of autophagy in pulmonary fibrosis was recently recognized. Several studies in IPF patients and BLM-challenged mice suggested that autophagy impairment contributes to the fibrosis response [45,46,47]. It was reported that the activation of lung fibroblasts is closely related to autophagy. Autophagy deficiency promotes myofibroblast differentiation and enhances the expression of α-SMA and FN in activated fibroblasts [48]. TJ-5 is a specific MyD88 inhibitor; Muhamuda Kader et al. demonstrated that MyD88 inhibits autophagy in liver macrophages to activate inflammasomes [26]. However, the regulatory role of TJ-5 in autophagy during lung fibrosis is unknown. Our study demonstrated that inhibiting MyD88 could trigger autophagy. LC3-II is closely related to autophagy, and p62 interacts with LC3 and is subsequently degraded; it accumulates in impaired autophagy [49]. Beclin-1 is the core component of the PI3K-III complex and regulates autophagy in several stages [50]. Based on these points, the amount of LC3-II, p62, and Beclin-1 reflects autophagic activity. TJ-5 increased the expression of LC3B-II and Beclin-1 and facilitated p62 degradation in experiments. Meanwhile, our results showed that the changes in TJ-5-autophagy were negatively correlated with the changes in α-SMA, FN, and COL1A1 proteins. Simultaneously, the protective effects of TJ-5 were abolished with the inhibition of autophagy by 3-MA, thus suggesting that the antifibrotic potential of TJ-5 may involve activation of autophagy.

Moreover, our results indicated that TJ-5 could induce apoptosis only at high concentrations (30 μmol/L). It was indeed reported that aberrant AKT activation promotes apoptosis resistance in senescent fibroblasts in IPF, and inhibition of AKT phosphorylation can induce apoptosis [51]. However, as shown in Figure 4E, the intervention of TJ-5 in the low dose range could not promote apoptosis of activated lung fibroblasts, and the effect on apoptosis was statistically significant when the concentration was increased to a high dose (30 μmol/L). To some extent, this result indicates that TJ-5 alleviates the apoptosis resistance of fibroblasts by inhibiting the phosphorylation of AKT is less obvious; that is, the pro-apoptotic effect of TJ-5 is less pronounced and only manifested at high doses. On the other hand, as shown in Figure 5E, TJ-5 could promote autophagy in activated fibroblasts when started at low doses (10 μmol/L). Compared with the effect of TJ-5 on apoptosis, the effect of TJ-5 on autophagy was more sensitive and pronounced.

Studies have confirmed that TLR4/MyD88 can affect autophagy activity by regulating mTOR in intestinal inflammation [23]. The mTOR signaling pathway, a major regulator of autophagy, may be regulated by PI3K/AKT and MAPK signaling pathways. For example, allicin attenuated myocardial hypertrophy by activating PI3K/AKT/mTOR and ERK/mTOR signaling [30]. Berberine induced autophagy in gastric cancer via the MAPK-dependent mTOR/p70S6K signaling pathway [52]. Furthermore, the PI3K/AKT and MAPK pathways also play a vital role in fibrotic diseases [53,54,55]. Suppression of PI3K/AKT/mTOR signaling alleviated pulmonary fibrosis in the BLM-induced animal model [56]. Inhibition of the ERK pathway prevented the progression of pulmonary fibrosis [57]. Collagen deposition in the fibrotic airway remodeling murine model and systemic profibrotic markers in IPF patients were all reduced with JNK inhibition [58]. Based on these findings, we aimed to determine whether MyD88 inhibition-induced autophagy in pulmonary fibrosis is related to mTOR, PI3K/AKT, and MAPK. Our results showed the activation of mTOR, PI3K/AKT, and MAPK signaling in BLM-induced fibrotic lung tissue and activated fibroblasts, and that TJ-5 effectively suppressed the phosphorylation of the mTOR, PI3K/AKT, and MAPK signaling pathways. These reassuring findings indicate that TJ-5 may regulate autophagy through the PI3K/AKT/mTOR and MAPK/mTOR signaling pathways, and TJ-5-induced autophagy may reduce pulmonary fibrosis. Overall, autophagy may be a critical target for TJ-5-mediated MyD88 inhibition, interfering with pulmonary fibrosis. It is worth mentioning that this is the first time that MyD88 has been shown to regulate autophagy in lung fibrosis.

It is worth noting that humans lacking functional MyD88 proteins are susceptible to a narrow range of pathogens, and only in infancy and early childhood. MyD88-dependent signaling appears to be dispensable for survival after adolescence [59]. The patients with MyD88 deficiency did not have serious and uncontrollable viral and fungal infections, and the bacterial infections in adulthood were also controllable, possibly because of immunological redundancies and an intact adaptive immune response [60]. In this study, by the end of our observation, short-term administration of TJ-5 did not increase the rate of infection in mice. However, in clinical infection lung disease, considering that TJ-5 may indeed promote infection due to its systemic MyD88 inhibition, according to clinical experience, antibiotics should be given concurrently for anti-infective treatment when TJ-5 is required to suppress inflammation in infectious diseases [61,62].

Furthermore, our previous studies have found that TJ-5 ameliorated renal interstitial fibrosis induced by ischemia–reperfusion injury by suppressing TGF-β1-induced epithelial-mesenchymal transition [63]. TJ-5 also alleviated myocardial fibrosis in mice after myocardial ischemia and reperfusion injury [64]. The inhibitory effect of TJ-5 on fibrosis in different organs further supports our results and suggests that MyD88 inhibition by TJ-5 may have a broad spectrum of anti-fibrosis effects. Moreover, we found no difference in serum ALT, AST, BUN, and Cr levels between mice treated with TJ-5 alone (30 mg/kg, intraperitoneal injection once a day for 21 days) and control mice (injection with an equal volume of ddH_2_O) (Appendix A), and liver and kidney tissues morphology were found to be normal in mice treated with TJ-5 alone (Appendix A). The results suggest that TJ-5 has no toxic side effects in mice. Previous studies have shown that the half-life of TJ-5 in the liver is very short, only 0.7 h, and there is no accumulation in the liver [65]. Combined with the fact that hepatotoxicity was not detected on the 21st day of the end point of pulmonary fibrosis efficacy observation in this study, it can be predicted that the safety of drugs in the treatment of pulmonary fibrosis and the risk of hepatotoxicity for longer use are small. Of course, it is also necessary to study the safety data of a longer dosing cycle before the clinic in the future.

Of course, this study has limitations. Toxicity assessment was limited to short-term studies of major organs, whereas comprehensive chronic toxicity studies are needed for a potential drug, including effects on the gastrointestinal tract, cardiovascular system, and hematopoiesis. The limitation of the BLM-induced mouse model is that it does not copy the characteristics of slow progress and irreversibility of human IPF. Although it can simulate the pathological characteristics of human IPF to a certain extent, it does not fully recapitulate the pathology of IPF in humans due to the differences among species [66]. These will be explored in future research for clinical applications.

## 5. Conclusions

In summary, the results of studies indicate that TJ-5 exerts a powerful protective effect on BLM-induced pulmonary fibrosis. This protection appears to be associated with the activation of autophagy induced by MyD88 inhibition. Further study reveals that TJ-5 may trigger autophagy by suppressing PI3K/AKT/mTOR and MAPK/mTOR signaling pathways. These findings provide a strong basis for the future clinical applications of TJ-5 for IPF.

## Figures and Tables

**Figure 1 biomedicines-13-02214-f001:**
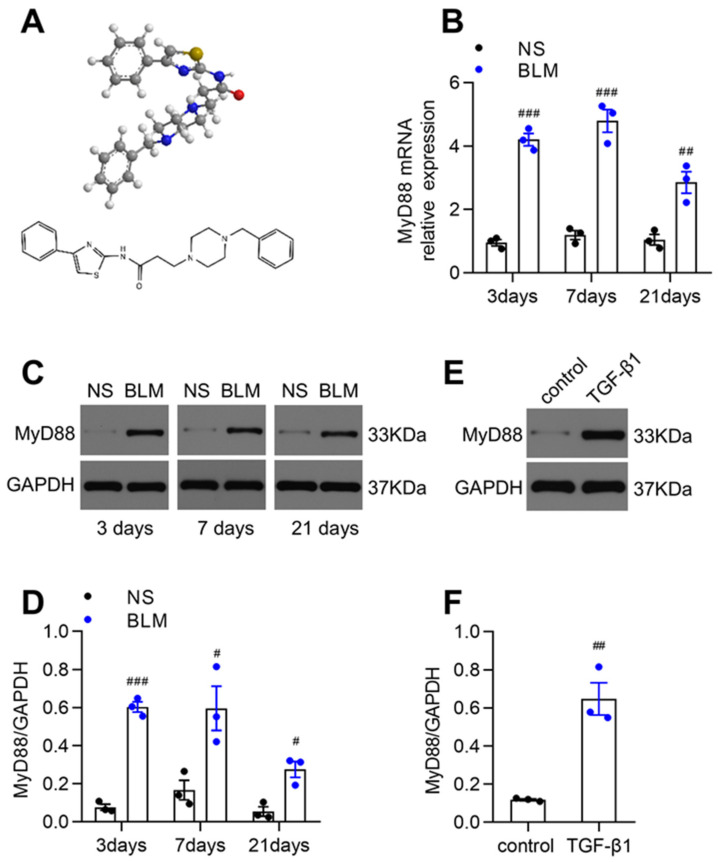
Increased MyD88 expression in BLM-challenged fibrotic lungs and TGF-β1-stimulated human lung fibroblasts. (**A**) The chemical structure of TJ-5. (**B**) Relative levels of MyD88 mRNA transcripts in lung tissues on days 3, 7, and 21 after the BLM challenge were quantified using RT-PCR. Normalization was performed with the housekeeping gene (GAPDH) (*n* = 3). ## *p* < 0.01 and ### *p* < 0.001 vs. the normal saline (NS) group. (**C**,**D**) Representative Western blot and quantitation of MyD88 in mice lungs on days 3, 7, and 21 after BLM challenge. GAPDH was used as the control (*n* = 3). # *p* < 0.05 and ### *p* < 0.001 vs. the NS group. (**E**,**F**) Representative Western blot and quantitation of MyD88 in MRC-5 human lung fibroblasts (*n* = 3). ## *p* < 0.01 vs. the control group. All data are presented as means ± SEM and are representative of three independent experiments.

**Figure 2 biomedicines-13-02214-f002:**
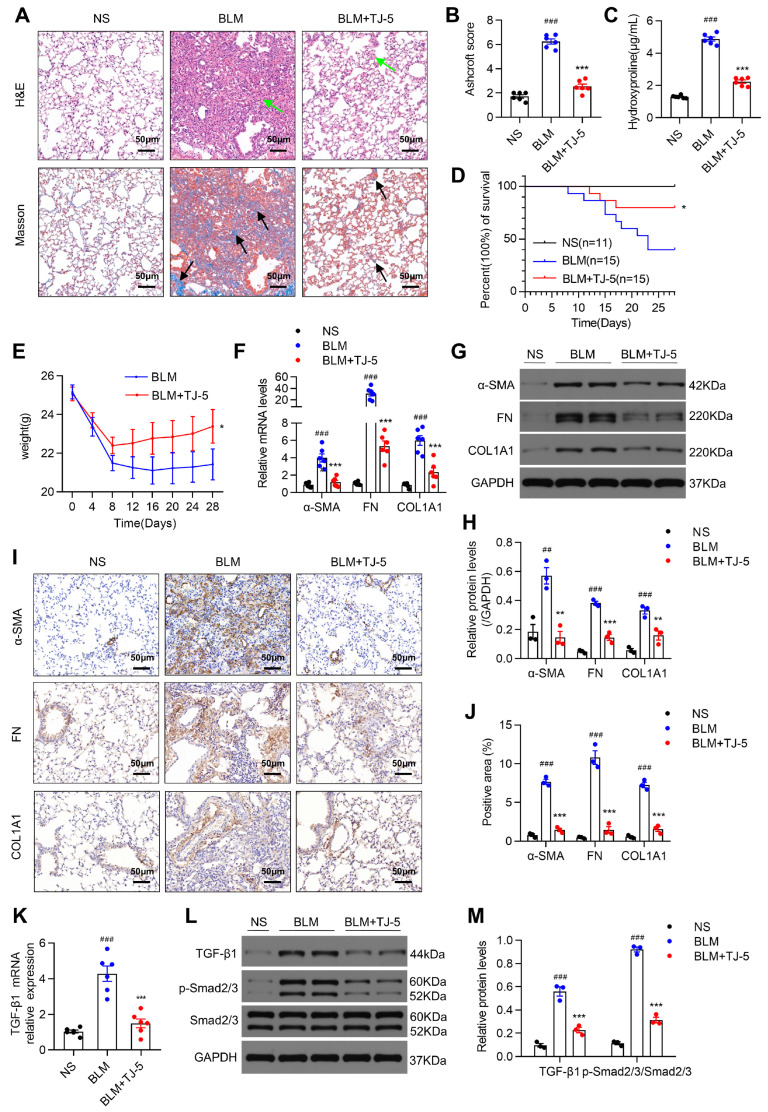
TJ-5 alleviates BLM-induced pulmonary fibrosis and regulates multiple fibrosis markers. (**A**) Hematoxylin and eosin (H&E) staining and Masson’s trichrome staining of lung sections at 21 days. Green arrows point to areas of fibrosis in the widened alveolar septum; black arrows point to blue areas indicative of collagen deposition. Scale bar: 50 μm. Evaluation of the severity of pulmonary fibrosis using the modified Ashcroft score (**B**) and hydroxyproline content (**C**) in lung tissues (*n* = 6). (**D**) After mice underwent the BLM challenge, survival was monitored for 28 days. The survival rate was 100% in the normal saline (NS) group, 40% in the BLM group, and 80% in the BLM + TJ-5 group (*n* = 10–15). (**E**) Bodyweight of the mice after BLM challenge for 28 days (*n* = 10–15). (**F**) Relative levels of α-SMA, FN, and COL1A1 mRNA transcripts in lung tissues on day 21 after BLM challenge (*n* = 6). (**G**,**H**) Representative Western blot and quantitation of α-SMA, FN, and COL1A1 in mice lungs on day 21 after BLM challenge (*n* = 3). (**I**,**J**) Immunohistochemical (IHC) staining and semi-quantitative analysis for α-SMA, FN, and COL1A1 in lung sections at 21 days. Scale bar: 50 μm. (**K**) Relative levels of TGF-β1 mRNA transcripts in lung tissues (*n* = 6). (**L**,**M**) Representative Western blot and quantitation of TGF-β1, p-Smad2/3, and Smad2/3 in mice lungs (*n* = 3). All data are presented as means ± SEM and are representative of three independent experiments. ## *p* < 0.01 and ### *p* < 0.001 vs. the normal saline (NS) group. * *p* < 0.05, ** *p* < 0.01, and *** *p* < 0.001 vs. the BLM group.

**Figure 3 biomedicines-13-02214-f003:**
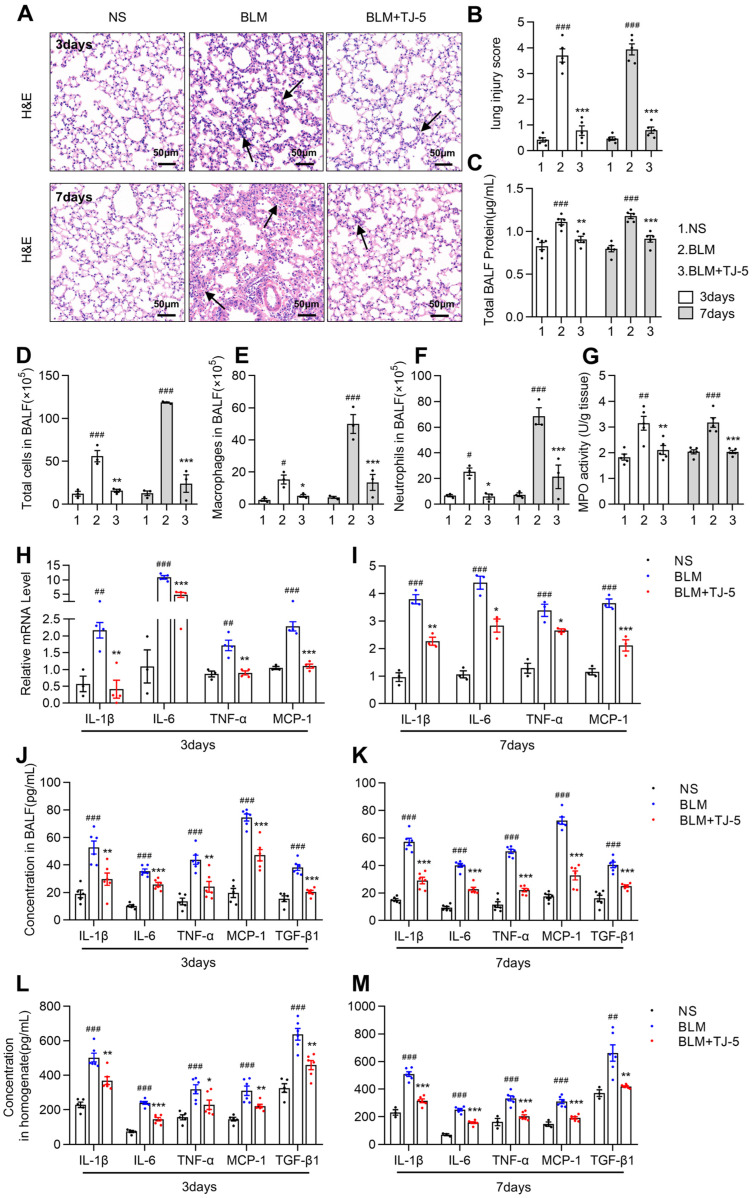
TJ-5 protects against the BLM-induced lung inflammatory response. Lung tissues and bronchoalveolar lavage fluid (BALF) were harvested on days 3 and 7 after BLM challenge. (**A**) Hematoxylin and eosin (H&E) staining of lung tissues. Black arrows point to areas with inflammatory infiltration in lung stroma. Scale bar: 50 μm. (**B**) Lung tissue injury was evaluated in all groups (*n* = 5). (**C**) Total protein concentration in BALF was detected using the BCA protein assay (*n* = 5). Total numbers of cells (**D**), macrophages (**E**), and neutrophils (**F**) in BALF (*n* = 3). (**G**) Myeloperoxidase (MPO) activity in lung tissue homogenate (*n* = 5). Relative levels of IL-1β, IL-6, TNF-α, and MCP-1 mRNA transcripts in lung tissues on days 3 (**H**) and 7 (**I**) were quantified using RT-PCR. Normalization was performed with the housekeeping gene (GAPDH) (*n* = 3–4). The protein expression levels of IL-1β, IL-6, TNF-α, MCP-1, and TGF-β1 in BALF on days 3 (**J**) and 7 (**K**) and in lung tissue homogenate on days 3 (**L**) and 7 (**M**) were measured using an ELISA assay (*n* = 3–6). All data are presented as means ± SEM and are representative of three independent experiments. # *p* < 0.05, ## *p* < 0.01, and ### *p* < 0.001 vs. the normal saline (NS) group. * *p* < 0.05, ** *p* < 0.01, and *** *p* < 0.001 vs. the BLM group.

**Figure 4 biomedicines-13-02214-f004:**
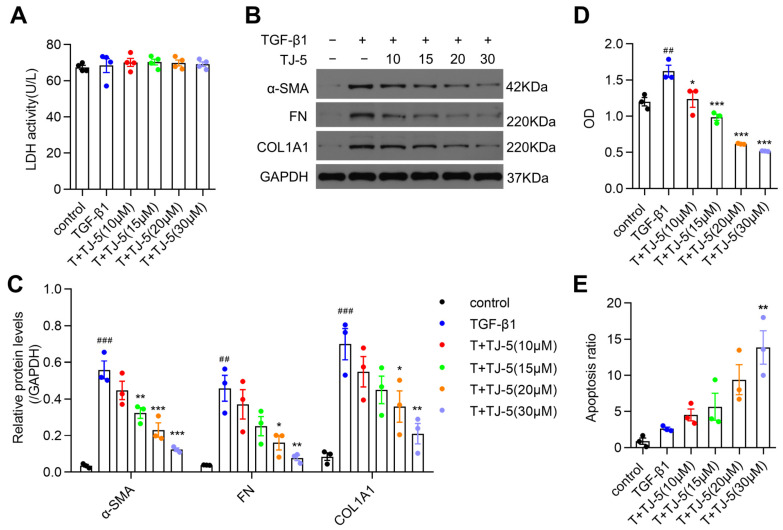
TJ-5 reduces TGF-β1-induced ECM deposition in lung fibroblasts and inhibits their proliferation. MRC-5 cells were pretreated with TJ-5 at varying concentrations (10 μM, 15 μM, 20 μM, 30 μM) for 2 h before being treated with TGF-β1 (5 ng/mL); 72 h later, cells were harvested. T + TJ-5 indicates TGF-β1 + TJ-5. (**A**) Cytotoxicity was measured using the LDH assay (*n* = 4). (**B**,**C**) Representative western blot and quantitation of α-SMA, FN, and COL1A1 in MRC-5 cells (*n* = 3). ## *p* < 0.01 and ### *p* < 0.001 vs. the control group. * *p* < 0.05, ** *p* < 0.01, and *** *p* < 0.001 vs. the TGF-β1 group. (**D**) The effect of TJ-5 on the proliferation of MRC-5 cells stimulated by TGF-β1 was measured using the CCK-8 assay (*n* = 3). ## *p* < 0.01 vs. the control group. * *p* < 0.05 and *** *p* < 0.001 vs. the TGF-β1 group. (**E**) Apoptosis of MRC-5 cells was detected using Annexin V/PI flow cytometry analysis (*n* = 3). ** *p* < 0.01 vs. the TGF-β1 group. All data are presented as means ± SEM and are representative of three independent experiments.

**Figure 5 biomedicines-13-02214-f005:**
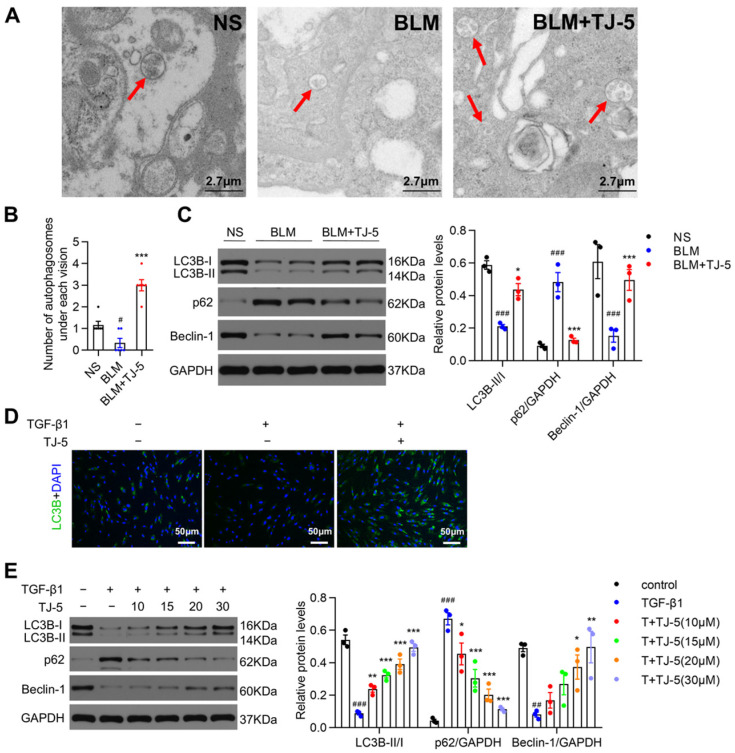
TJ-5 activates autophagy in lung tissue. (**A**,**B**) The number of autophagosomes in lung tissues on day 21 after BLM challenge was detected by transmission electron microscopy (80 kV, 5000×; red arrows indicate autophagosomes) (*n* = 6). # *p* < 0.05 vs. the normal saline (NS) group. *** *p* < 0.001 vs. the BLM group. (**C**) Representative Western blot and quantitation of LC3B, p62, and Beclin-1 in mice lungs on day 21 after BLM challenge (*n* = 3). ### *p* < 0.001 vs. the NS group. * *p* < 0.05 and *** *p* < 0.001 vs. the BLM group. (**D**) Punctate LC3B staining was measured by immunofluorescence in MRC-5 cells. Punctate LC3B staining is shown in green. Scale bar: 50 μm. (**E**) Representative Western blot and quantitation of LC3B, p62, and Beclin-1 in MRC-5 cells (*n* = 3). ## *p* < 0.01 and ### *p* < 0.001 vs. the control group. * *p* < 0.05, ** *p* < 0.01, and *** *p* < 0.001 vs. the TGF-β1 group. All data are presented as means ± SEM and are representative of three independent experiments.

**Figure 6 biomedicines-13-02214-f006:**
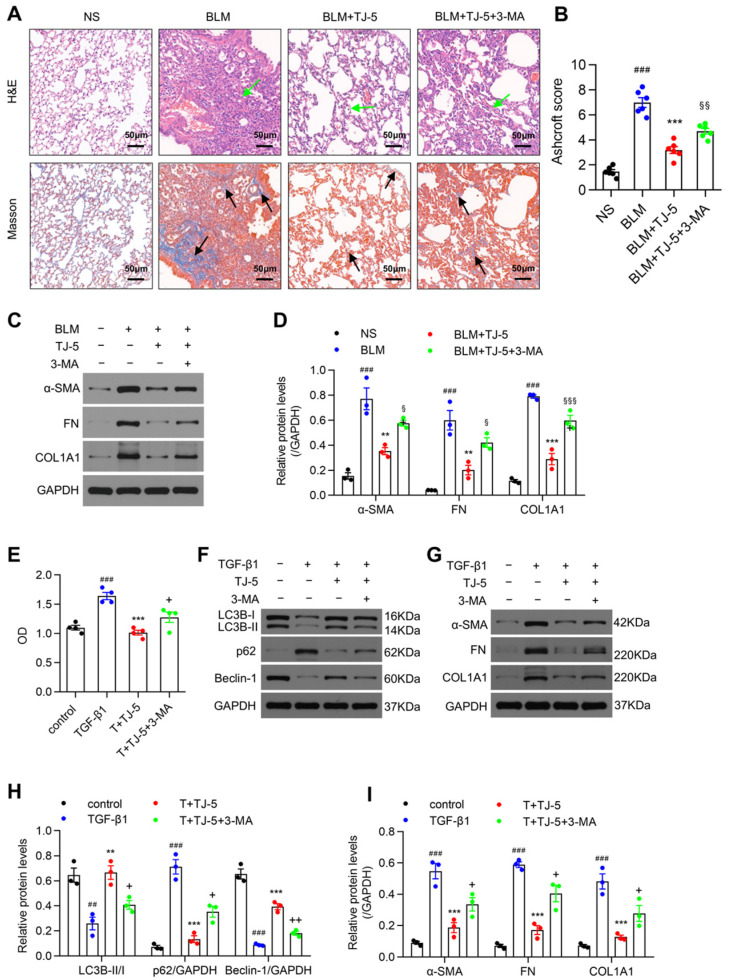
Inhibition of autophagy reverses the positive effects of TJ-5. The BLM + TJ-5 + 3-MA group was intraperitoneally injected with TJ-5 (30 mg/kg/day) and 3-MA (15 mg/kg/day) for three weeks. (**A**) Hematoxylin and eosin (H&E) staining and Masson’s trichrome staining of lung sections at 21 days. Green arrows point to areas of fibrosis in the widened alveolar septum; black arrows point to blue areas indicative of collagen deposition. Scale bar: 50 μm. (**B**) Evaluation of the severity of pulmonary fibrosis using the modified Ashcroft score in lung tissues (*n* = 6). (**C**,**D**) Representative western blot and quantitation of α-SMA, FN, and COL1A1 in mice lungs on day 21 after BLM challenge (*n* = 3). ### *p* < 0.001 vs. the normal saline (NS) group. ** *p* < 0.01 and *** *p* < 0.001 vs. the BLM group. ^§^
*p* < 0.05, ^§§^
*p* < 0.01, and ^§§§^
*p* < 0.001 vs. the BLM + TJ-5 group. MRC-5 cells were pretreated with TJ-5 at a concentration of 20 μM for 2 h, pretreated or not pretreated with 3-MA (2.5 mM) for 4 h, and then treated with TGF-β1 (5 ng/mL) for 72 h. T + TJ-5 indicates TGF-β1 + TJ-5. (**E**) The proliferation of MRC-5 cells was measured using the CCK-8 assay (*n* = 4). ### *p* < 0.001 vs. the control group. *** *p* < 0.001 vs. the TGF-β1 group. **^+^**
*p* < 0.05 vs. the T + TJ-5 group. Representative western blot and quantitation of LC3B, p62, Beclin-1 (**F**,**H**), α-SMA, FN, and COL1A1 (**G**,**I**) in MRC-5 cells (*n* = 3). ## *p* < 0.01 and ### *p* < 0.001 vs. the control group. ** *p* < 0.01 and *** *p* < 0.001 vs. the TGF-β1 group. **^+^*** p* < 0.05 and ^++^
*p* < 0.01 vs. the TGF-β1 + TJ-5 group. All data are presented as means ± SEM and are representative of three independent experiments.

**Figure 7 biomedicines-13-02214-f007:**
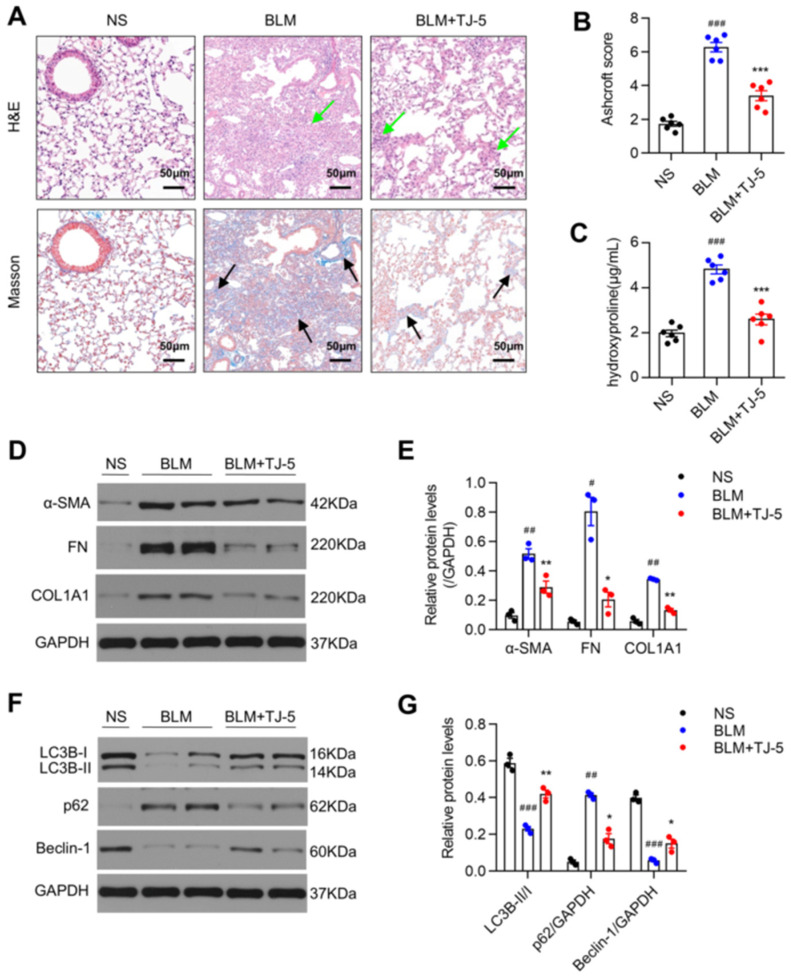
Post-treatment with TJ-5 alleviates BLM-induced pulmonary fibrosis. TJ-5 was administered (30 mg/kg/day, intraperitoneal) at D7 after BLM injection, and the lungs were removed at D21 after BLM or saline administration. (**A**) Hematoxylin and eosin (H&E) staining and Masson’s trichrome staining of lung sections at 21 days. Green arrows point to areas of fibrosis in the widened alveolar septum; black arrows point to blue areas indicative of collagen deposition. Scale bar: 50 μm. Evaluation of the severity of pulmonary fibrosis using the modified Ashcroft score (**B**) and hydroxyproline content (**C**) in lung tissues (*n* = 6). (**D**,**E**) Representative Western blot and quantitation of α-SMA, FN, and COL1A1 in mice lungs on day 21 after BLM challenge (*n* = 3). (**F**,**G**) Representative Western blot and quantitation of LC3B, p62, and Beclin-1 in mice lungs (*n* = 3). All data are presented as means ± SEM and are representative of three independent experiments. # *p* < 0.05, ## *p* < 0.01, and ### *p* < 0.001 vs. the normal saline (NS) group. * *p* < 0.05, ** *p* < 0.01, and *** *p* < 0.001 vs. the BLM group.

**Figure 8 biomedicines-13-02214-f008:**
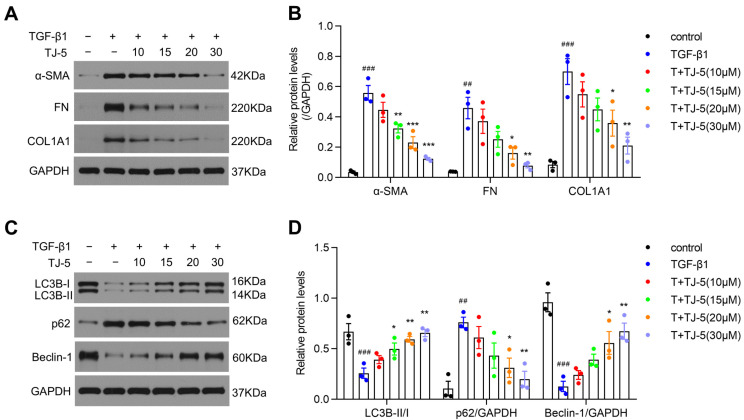
Post-treatment with TJ-5 inhibited lung fibroblast activation and induced autophagy. MRC-5 fibroblasts were first stimulated by TGF-β for 4 h, followed by TJ-5 stimulation at varying concentrations (10 μM, 15 μM, 20 μM, 30 μM) 72 h later, then cells were harvested. T + TJ-5 indicates TGF-β1 + TJ-5. (**A**,**B**) Representative Western blot and quantitation of α-SMA, FN, and COL1A1 in MRC-5 cells (*n* = 3). (**C**,**D**) Representative Western blot and quantitation of LC3B, p62, and Beclin-1 in MRC-5 cells (*n* = 3). ## *p* < 0.01 and ### *p* < 0.001 vs. the control group. * *p* < 0.05, ** *p* < 0.01, and *** *p* < 0.001 vs. the TGF-β1 group.

**Figure 9 biomedicines-13-02214-f009:**
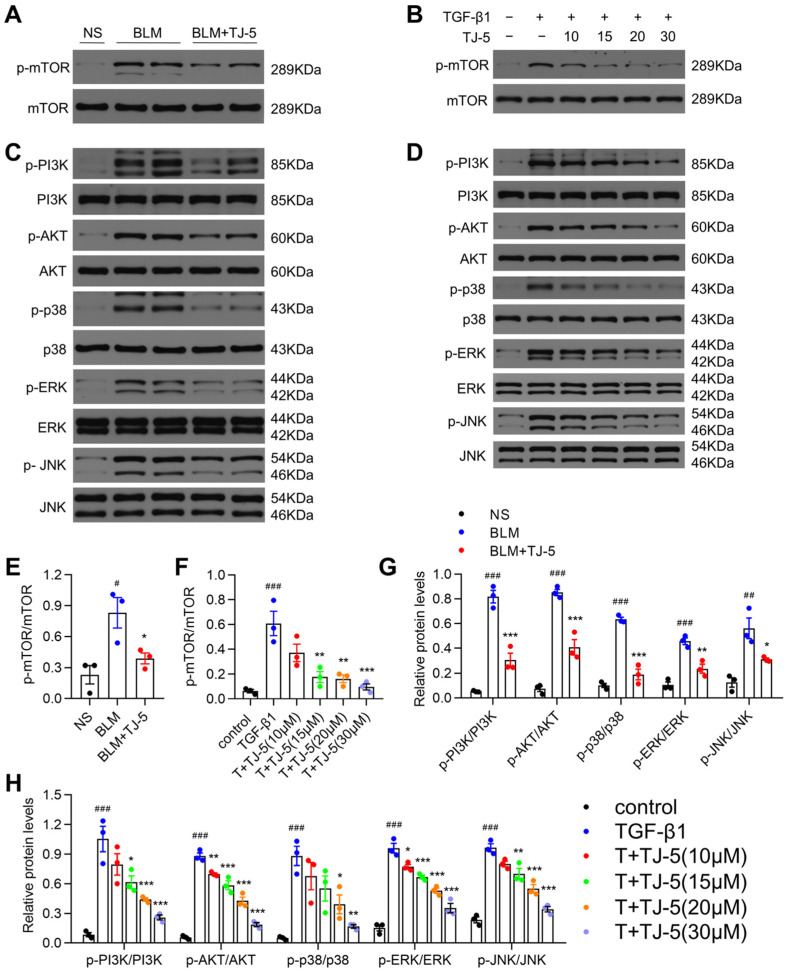
TJ-5-mediated MyD88 inhibition triggers autophagy via PI3K/AKT/mTOR and MAPK/mTOR signaling pathways. Representative western blot and quantitation of mTOR and its phosphorylation, p-mTOR in lung tissues (**A**,**E**), and MRC-5 cells (**B**,**F**) (*n* = 3). # *p* < 0.05 and ### *p* < 0.001 vs. the normal saline (NS) group or control group. * *p* < 0.05, ** *p* < 0.01, and *** *p* < 0.001 vs. the BLM group or TGF-β1 group. Representative Western blot and quantitation of p-PI3K, PI3K, p-AKT, AKT, p-p38, p38, p-ERK, ERK, p-JNK, and JNK in lung tissues (**C**,**G**) and MRC-5 cells (**D**,**H**) (*n* = 3). ## *p* < 0.01 and ### *p* < 0.001 vs. the NS group or control group. * *p* < 0.05, ** *p* < 0.01 and *** *p* < 0.001 vs. the BLM group or TGF-β1 group. All data are presented as means ± SEM and are representative of three independent experiments.

## Data Availability

The data presented in this study are available on request from the corresponding authors.

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
