# Peer review of "Attenuation of Pulmonary Fibrosis by the MyD88 Inhibitor TJ-M2010-5 Through Autophagy Induction in Mice"

_biomedicines, 2025, doi:10.3390/biomedicines13092214_

Round 1

Reviewer 1 Report

Comments and Suggestions for Authors

The manuscript entitled “Attenuation of Pulmonary Fibrosis by the MyD88 Inhibitor TJ- 2M2010-5 through Autophagy Induction in Mice” has been reviewed.

The authors have considered and addressed various aspects in examining their hypothesis. The manuscript is well-prepared, with appropriate methodologies. The results were clear and well-organized. The discussion was also comprehensive. However, a few comments are given as follows:

Introduction

- It would be better to mention the prevalence of Idiopathic pulmonary fibrosis.

- Page 2, line 52-53, only one study (22) was cited for the following sentence: “Meanwhile, several studies have found that MyD88 is closely associated with autophagy”. Given that the term "several studies" is used, it would be better to refer to several other relevant cases.

Materials and Methods

-It would be better to mention the total number of mice used in the study and in each group in the section of Materials and Methods (page 3, 2.2 Animals and Animal Treatment).

- It would be better to explain a little about how animals are finally sacrificed.

Results

- Page 8, line 336, there appears to be a typo in “Autophagy activaty”. Please correct it to “activity”.

Discussion

- Please add a relevant reference for the following statement: “In addition, in the clinical infection lung disease, considering that TJ-5 may indeed promote infection due to its systemic MyD88 inhibition, according to clinical experience, antibiotics should be given concurrently for anti-infective treatment when TJ-5 is required to suppress inflammation in infectious diseases.” (Page 21, line 561-564)

References

- The list of references seems to be missing in the manuscript.

Author Response

Dear Reviewer:

Thank you very much for your comments! Revised portions are marked in red in the revised manuscript.

1st comment: It would be better to mention the prevalence of Idiopathic pulmonary fibrosis.

Response to 1st comment: Thank you for your kind suggestion. We added [The global incidence and prevalence of IPF are estimated to be in the range of 0.09–1.30 and 0.33–4.51 per 10,000 persons, respectively] in line 41-42 to mention the prevalence of Idiopathic pulmonary fibrosis (doi: 10.1186/s12931-021-01791-z).

2nd comment: Page 2, line 52-53, only one study (22) was cited for the following sentence: “Meanwhile, several studies have found that MyD88 is closely associated with autophagy”. Given that the term "several studies" is used, it would be better to refer to several other relevant cases.

Response to 2nd comment: Thanks for your comment. We cited additional two documents (doi: 10.3390/ijms24054979 and 10.1016/j.cell.2017.07.008) in line 61-62 to clearly describe MyD88 is closely associated with autophagy.

3rd comment: It would be better to mention the total number of mice used in the study and in each group in the section of Materials and Methods (page 3, 2.2 Animals and Animal Treatment).

Response to 3rd comment: Thanks for your suggestion. We added [A total of 54 mice were used, with 6 mice in each group.] in line 104 to mention the total number of mice used in the study and in each group.

4th comment: It would be better to explain a little about how animals are finally sacrificed.

Response to 4th comment: Thanks for your kind comment. We added [Mice were sacrificed by cervical dislocation under deep anesthesia at the end of the experiment] in line 118-119 to explain a little about how animals are finally sacrificed.

5th comment: Page 8, line 336, there appears to be a typo in “Autophagy activaty”. Please correct it to “activity”.

Response to 5th comment: Thanks for your comment. We corrected "autophagy activaty " to "autophagy activity" in line 350.

6th comment: Please add a relevant reference for the following statement: “In addition, in the clinical infection lung disease, considering that TJ-5 may indeed promote infection due to its systemic MyD88 inhibition, according to clinical experience, antibiotics should be given concurrently for anti-infective treatment when TJ-5 is required to suppress inflammation in infectious diseases.” (Page 21, line 561-564)

Response to 6th comment: Thanks for your kind comment. We added the relevant reference (doi: 10.1097/MD.0b013e3181fd8ec3 and 10.1126/science.1158298) in line 586.

7th comment: The list of references seems to be missing in the manuscript.

Response to 7th comment: We are sorry that the list of references seems to be missing in the manuscript. The list of reference is in page 22-26 of Word manuscript.

Thank you very much for your comments again. After we revised manuscript according to your suggestion, we felt that the article was more logical.

Best wishes.

Reviewer 2 Report

Comments and Suggestions for Authors

1) Although the study convincingly demonstrates that the MyD88 inhibitor TJ-5 effectively suppresses pulmonary fibrosis via activation of autophagy, the data obtained raise serious concerns regarding its clinical applicability. A key issue is the insufficient study of safety issues, especially given the central role of MyD88 in immune defense. The authors did not provide data on the effect of the drug on resistance to infections, which is critically important since patients with IPF are already susceptible to pulmonary infections. Toxicity assessment was limited to short-term studies of major organs, whereas comprehensive chronic toxicity studies are needed for a potential drug, including effects on the gastrointestinal tract, cardiovascular system, and hematopoiesis. Also missing is an analysis of possible compensatory activation of alternative inflammatory pathways with long-term use. These gaps significantly limit the translational potential of the work. At a minimum, an in-depth discussion of this limitation is necessary

2) The bleomycin mouse model does not fully recapitulate the pathogenesis of IPF in humans. The limitations of the model should be clearly stated.

3) The results of the study show that the ability of TJ-5 to induce cell apoptosis is observed only at the maximum tested doses (30 μM), while in the clinically significant concentration range (10-20 μM) this effect is practically absent. Such data cast doubt on the fact that apoptosis induction plays a significant role in the mechanism of the antifibrotic action of the compound. Particularly noteworthy is the fact that the work does not present an in-depth analysis of the molecular mechanisms of this phenomenon. In particular: key biomarkers of the apoptotic cascade (the level of caspase activation, changes in the expression of Bcl-2 family proteins) were not studied; possible alternative forms of programmed cell death were not considered. More direct measurements of apoptosis should be provided

4) The authors tested only short-term effects (21 days) on the liver and kidneys. Long-term study data or at least a discussion of the varmacokinetics and metabolism of TJ-5 (e.g. the risk of hepatotoxicity with chronic use) are needed.

5) Graphic abstract is required

Author Response

Dear Reviewer:

Thank you very much for your comments! Revised portions are marked in red in the revised manuscript.

1st comment: Although the study convincingly demonstrates that the MyD88 inhibitor TJ-5 effectively suppresses pulmonary fibrosis via activation of autophagy, the data obtained raise serious concerns regarding its clinical applicability. A key issue is the insufficient study of safety issues, especially given the central role of MyD88 in immune defense. The authors did not provide data on the effect of the drug on resistance to infections, which is critically important since patients with IPF are already susceptible to pulmonary infections. Toxicity assessment was limited to short-term studies of major organs, whereas comprehensive chronic toxicity studies are needed for a potential drug, including effects on the gastrointestinal tract, cardiovascular system, and hematopoiesis. Also missing is an analysis of possible compensatory activation of alternative inflammatory pathways with long-term use. These gaps significantly limit the translational potential of the work. At a minimum, an in-depth discussion of this limitation is necessary.

Response to 1st comment: Thank you for your constructive suggestion. We added [It is worth noting that humans lacking functional MyD88 proteins are susceptible to a narrow range of pathogens, and only in infancy and early childhood. MyD88 de-pendent signaling appears to be dispensable for survival after adolescence [59]. The patients with MyD88 deficiency did not have serious and uncontrollable viral and fungal infections, and the bacterial infections in adulthood were also controllable, possibly because of immunological redundancies and an intact adaptive immune re-sponse [60]. In this study, by the end of our observation, short-term administration of TJ-5 did not increase the rate of infection in mice.] in line 576-583 (doi: 10.1002/eji.201242683 and 10.1097/TP.0000000000004725) to discuss the effect of the MyD88 inhibition on resistance to infections. We added [Of course, this study has limitations. Toxicity assessment was limited to short-term studies of major organs, whereas comprehensive chronic toxicity studies are needed for a potential drug, including effects on the gastrointestinal tract, cardiovascular system, and hematopoiesis.] in line 604-606 to discussion the limitation of safety study.

2nd comment: The bleomycin mouse model does not fully recapitulate the pathogenesis of IPF in humans. The limitations of the model should be clearly stated.

Response to 2nd comment: Thanks for your suggestion. This is necessary. We added [The limitation of the BLM-induced mouse model is that it does not copy the charac-teristics of slow progress and irreversibility of human IPF. Although it can simulate the pathological characteristics of human IPF to a certain extent, it does not fully recapit-ulate the pathology of IPF in humans due to the differences among species [66]. These will be explored in future research for clinical applications] in line 607-611 (doi: 10.1038/s41588-024-01819-2).

3rd comment: The results of the study show that the ability of TJ-5 to induce cell apoptosis is observed only at the maximum tested doses (30 μM), while in the clinically significant concentration range (10-20 μM) this effect is practically absent. Such data cast doubt on the fact that apoptosis induction plays a significant role in the mechanism of the antifibrotic action of the compound. Particularly noteworthy is the fact that the work does not present an in-depth analysis of the molecular mechanisms of this phenomenon. In particular: key biomarkers of the apoptotic cascade (the level of caspase activation, changes in the expression of Bcl-2 family proteins) were not studied; possible alternative forms of programmed cell death were not considered. More direct measurements of apoptosis should be provided

Response to 3rd comment: Thanks for your insightful suggestion. Cleaved-caspase 3 and Bcl-2 expression have been detected to analyze the molecular mechanisms of apoptosis. We added [Furthermore, we detected the expressions of Cleaved-caspase 3 and Bcl-2, and found that TJ-5 at 20 μM and 30 μM markedly downregulated cleaved caspase-3 levels, while upregulating Bcl-2 (Figure S2).] in line 307-311. And we can observe that apoptosis level slowly and gradually increased as the concentration of TJ-5 increased, which is in line with the fact that apoptosis induction plays a significant role in the mechanism of the antifibrotic action. The data is in Supplemental Figure S2 of the revised manuscript, as follows:

Supplementary Fig. S2. The effect of TJ-5 on the apoptosis of TGF-β1-induced MRC-5 cells. MRC-5 cells were pretreated with TJ-5 at varying concentrations (10 μM, 15 μM, 20 μM, 30 μM) for 2 h before being treated with TGF-β1 (5 ng/mL); 72 hours later, cells were harvested. T+TJ-5 indicates TGF-β1+TJ-5. (A) Representative western blot and quantitation of Cleaved-caspase-3 and Bcl-2 in MRC-5 cells (n = 3). **P < 0.01, ***P < 0.001 vs the TGF-β1 group. All data are presented as means ± SEM and are representative of three independent experiments.

4th comment: The authors tested only short-term effects (21 days) on the liver and kidneys. Long-term study data or at least a discussion of the varmacokinetics and metabolism of TJ-5 (e.g. the risk of hepatotoxicity with chronic use) are needed.

Response to 4th comment: Thanks for your suggestion. We added [Previous studies have shown that the half-life of TJ-5 in the liver is very short, only 0.7 h, and there is no accumulation in the liver [65]. Combined with the fact that hepatotoxicity was not detected on the 21st day of the end point of pulmonary fibrosis efficacy observation in this study, it can be predicted that the safety of drugs in the treatment of pulmonary fibrosis and the risk of hepatotoxicity for longer use are small. Of course, it is also necessary to study the safety data of longer dosing cycle before clinic in the future] in line 597-603 to discuss the safety data and the varmacokinetics of TJ-5 (doi: 10.3389/fphar.2022.1080438).

5th comment: Graphic abstract is required.

Response to 5th comment: We are sorry that Graphic abstract seems to be missing. We provided the Graphic abstract in Supplementary_Material, as follows:

Thank you very much for your comments again. After we revised manuscript according to your suggestion, we felt that the article was more logical.

Best wishes.

Round 2

Reviewer 2 Report

Comments and Suggestions for Authors

The authors have done a great job and have significantly improved the article. The authors have managed to take into account all my comments. The article can be accepted for publication in its current form.